**Data Availability Statement:** We've made our minimal anonymized data set publicly available on

# Anorectal Malformations (ARM) and associated maternal factors among children at Tikur Anbessa Specialized Hospital and St. Paul's Hospital Millennium Medical College, Addis Ababa, Ethiopia: An unmatched case-control study

Samrawit Solomon[1], Fisseha Temesgen[2], Solomon Tibebu[3]*, Hana Abebe[4], Girma Seyoum[3]

1 Department of Anatomy, School of Medicine, College of Health Sciences and Medicine, Dilla University, Dilla, Ethiopia, 2 Department of Surgery, School of Medicine, College of Health Sciences, Addis Ababa University, Addis Ababa, Ethiopia, 3 Department of Anatomy, School of Medicine, College of Health Sciences, Addis Ababa University, Addis Ababa, Ethiopia, 4 Department of Surgery, School of Medicine, St. Paul's Hospital Millennium Medical College, Addis Ababa, Ethiopia

* solomon.tibebu@aau.edu.et

## Abstract

### Introduction

Anorectal malformations (ARMs) are birth defects that affect the rectum, anus, and surrounding structures. While genetic and environmental factors may contribute to the risk of developing ARMs, the exact causes are largely unknown. Notably, there is a lack of research investigating predisposing factors for ARMs within the Ethiopian population, despite the burden of this condition in the country.

### Objective

The research study aimed at to examine the maternal risk factors linked to the occurrence of anorectal abnormalities in children receiving treatment at designated public hospitals located in Addis Ababa, Ethiopia.

### Methods

An unmatched case-control study was conducted at selected hospitals on mothers and their children between August 2022 and January 2023. The sample consisted of pediatric patients admitted to pediatric surgical units. Cases were diagnosed with ARMs, while controls had no congenital anomalies. Data was collected from the mothers of both cases and controls. The data was validated and then exported to SPSS version 26 for analysis. The analysis employed descriptive statistics and binary logistic regression. In a multivariable

the Open Science Framework (OSF), a recommended platform for sharing research data. You can find the data set at the following URL: [https://osf.io/nrqkx/].

**Funding:** The author(s) received no specific funding for this work.

**Competing interests:** The authors have declared that no competing interests exist.

model, an adjusted odds ratio (AOR) together with a 95% confidence interval and p-value < 0.05 was used to determine significance.

## Result

This study included 68 ARM cases and 136 controls. Multivariable analysis found that a family history of birth defects (AOR = 6.15, 95% CI: 1.24–30.58), maternal alcohol use (AOR = 4.71, 95% CI: 1.71–12.00), first-trimester medication use (AOR = 4.86, 95% CI: 1.29–18.32), advanced maternal age (AOR = 4.22, 95% CI: 1.21–14.69), and unplanned pregnancy (AOR = 3.701, 95% CI: 1.551–8.828) were significant risk factors for ARM.

## Conclusion

The study found that key risk factors for ARM include family history of birth defects, maternal alcohol use, first-trimester medication use, advanced maternal age, and unplanned pregnancy. These findings underscore the importance of tailored prevention strategies and screening programs to address the genetic, maternal lifestyle, and maternal health factors that contribute to this congenital disorder.

## Introduction

Anorectal malformation (ARM) is a congenital anomaly that encompasses a wide spectrum of defects involving the anus and rectum, along with the genitourinary system in both sexes. These defects range from minor skin-level issues, such as rectoperineal or anoperineal fistulas, to more complicated defects with associated anomalies [1–4]. The birth prevalence of ARM ranges from 1 in 1500 to 1 in 5000 live births worldwide, with geographical and seasonal variations [4]. Its birth prevalence is 3.59 per 10,000 in Germany and 3.09 per 10,000 in Italy [1, 5]. In South Africa, the prevalence ranges from 1.79 to 3.06 per 10,000, with a male predominance [2].

During normal human embryology, the terminal part of the hindgut forms a structure called the cloaca. Around the seventh week of development, the cloaca splits by the urorectal septum, creating the ventral opening of the urogenital sinus and the anal opening of the hindgut. Most anorectal anomalies result from incomplete separation of the cloaca into urogenital and anorectal portions by the urogenital septum [6]. Lesions of ARM are classified as low or high depending on whether the rectum ends superior or inferior to the pubo-rectalis muscle. The low anomalies of the anorectal region include Anal Agenesis, with or without a Fistula, Anal Stenosis, membranous Atresia of Anus, Anorectal Agenesis, with or without a Fistula. The high anomalies include recto-vesical fistulas, rectovaginal fistulas and recto-bladder neck fistula [3, 6].

According to the new krickenbeck classification system anorectal malformation is classified as major clinical group and rare variation. The major clinical group is further classified as perineal fistula, anal stenosis, recto-urethral fistula, cloacal malformation, recto vesical fistula. The rare variants are sub-classified as rectal stenosis, rectovaginal fistula, H-type fistula and pouch colon [7]. The most frequent ARM subtype in male is recto-urethral fistula and vestibular fistula in female. About 48%-78% ARM are associated with other anomalies [8, 9].

The etiology of anorectal malformations (ARMs) are the subject of ongoing research and debate. The tendency to develop ARMs is believed to result from the combined influence of various genetic and environmental factors, rather than being due to a single causal element.

While genetic factors have been associated with ARMs, the precise genetic mechanisms are not fully understood. In rare syndromic types of ARMs, such as Townes–Brocks syndrome, Down syndrome, and Currarino syndrome, the cause can be explained solely by genetic factors, such as mutations in specific genes and chromosomal structural abnormalities [3]. However, for the majority of ARMs, the underlying genetic basis is more complex. Animal experiments have shown that problems in the *SHH* (sonic hedgehog signaling molecule) and *TCF4* (transcription factor 4) genes, which are important in the development of the digestive tract during human embryology, can result in ARMs [10].

However in non-syndromic types, in addition to genetic factors, environmental risk factors have also been identified. These include maternal smoking [11], maternal obesity [12], maternal diabetes [12], maternal overweight [13], previous miscarriage [14], and maternal chronic respiratory disease [15]. Understanding the full range of predisposing factors, both genetic and environmental, will be crucial for developing comprehensive strategies for the prevention and management of anorectal malformations.

ARMs represent a significant burden for pediatric surgery, both in developed and developing countries. Literature from Africa in particular has distinguished ARMs as the leading congenital anomaly and the most common cause of neonatal gastrointestinal obstruction among congenital pediatric surgery cases [16–18]. Research conducted in Nigeria and Uganda found that ARMs account for a substantial proportion of congenital anomaly cases, at 21.2% and 19% respectively. Additionally, ARMs were the leading cause of intestinal obstruction, responsible for 57–67% of such cases in these countries [2, 16, 17, 19]. These findings underscore the disproportionate impact of ARMs on pediatric surgical caseloads, especially in resource-constrained settings. Addressing this public health challenge will require concerted efforts to improve early diagnosis, access to specialized surgical care, and long-term management strategies for affected children and their families.

While there is limited research on the birth prevalence and incidence of ARMs in Ethiopia, hospital based studies have highlighted the substantial burden of this condition on the country's pediatric surgery units. A study conducted at Tikur Anbessa Specialized Hospital (TASH) from 2010 to 2014 found that gastrointestinal tract (GIT) problems accounted for 43.3% of all neonatal surgical cases, with ARMs representing 19% of these GIT-related congenital anomalies and 24% of all neonatal surgical cases [20].

Furthermore, a separate five-year retrospective study at TASH identified ARMs as the first of the top ten congenital anomalies (8.9%) requiring surgical intervention, and the second most common cause of pediatric admissions after foreign body swallowing [21]. In addition, ARMs were also shown to be the most common cause of intestinal obstruction in the early new-born period, accounting for 57% of cases, followed by intestinal atresia (13%) and Hirsschsprung's disease (12%) in a study conducted at the TASH neonatal care unit [22]. Furthermore, a study at Jimma Medical Centre between 2017 and 2019 found that ARMs, primarily imperforate anus, were the third most common congenital anomaly (9%), preceded only by spinal bifida (14.6%) and clubfoot (12.56%) [23].

These findings underscore the significant impact of ARMs on pediatric surgical caseloads in Ethiopia. The high burden ARMs in this population highlights the pressing need for increased research efforts to better understand the underlying causes, risk factors, long-term outcomes, and prevention strategies of these congenital anomalies.

Despite the substantial burden of ARMs, there is a paucity of literature quantifying the determinants or risk factors associated with the development of this condition in Ethiopia or across the African continent. Understanding the specific maternal risk factors could aid in preparing prenatal caregivers for early recognition and timely interventions. This knowledge could allow healthcare providers to counsel women on modifiable risk factors, optimize

antenatal care, implement targeted interventions during pregnancy, and inform strategies for prevention. To address this gap, the present study aims to identify potential risk factors associated with the development of ARM.

## Method

### Study area and period

This study was conducted at Tikur Anbessa Specialized Hospital (TASH) and St Paul's Hospital Millennium Medical College (SPHMMC) pediatric Surgical units from August 2022-January 2023 G.C. TASH is Ethiopia's largest territory hospital, by far. It provides comprehensive medical services to both the local population and referred patients from other areas. Moreover, TASH stands out as one of the few specialized hospitals in the country that specifically caters to pediatric surgical needs. Located in the northwestern part of Addis Ababa, SPHMMC is a tertiary hospital in the city. The Department of Surgery at SPHMMC is organized into three units, with the pediatric surgery unit being one of them.

### Study design

A hospital-based unmatched case-control study was conducted, where the participants included pediatric patients admitted to the pediatric surgical units with a diagnosis of ARM as cases, and pediatric patients without any congenital anomaly who sought treatment of other illnesses during the study period as controls.

### Source population

All children with anorectal malformation and their mothers who received care at Tikur Anbessa Specialized Hospital and St Paul's Hospital Millennium Medical College in Addis Ababa, Ethiopia during the study period of August 2022 to January 2023.

### Study population

This includes Children with ARM (cases) and children without the condition (controls), along with their mothers, who were selected from the source population at the same two hospitals during the time period between August 2022 and January 2023).

### Study unit

All mothers of both control and case groups.

### Diagnosis of ARM

Patients were diagnosed with ARM through a two-step process. Pediatric surgeons first conducted a clinical examination to identify the characteristic features. If the clinical findings were unclear, a perineal ultrasound was performed to confirm the diagnosis. Only patients with a definitive ARM diagnosis confirmed by this combined clinical and imaging approach were included in the study.

### Inclusion/exclusion criteria

Children diagnosed with anorectal malformations (ARM) with the above diagnostic approach and admitted with their biological mothers were included as the case group. However, the study excluded any children admitted with a caregiver other than their biological mother, for

both the case and control groups. Furthermore, the control group excluded any children diagnosed with other congenital anomalies.

## Sample size determination

Sample size was calculated using the Epi Info 7 statistical software package. The study used a two-sided significance level of 95%, a power of 80%, a control-to-case ratio of 2:1, and an estimated exposure status of 7.2% for cases and 0.2% for controls [13]. Since the data collection involved retrospective chart reviews and interviews with mothers of children born with ARM, rather than long-term participant follow-up, a 5% non-response rate was determined to be adequate for the study. With these considerations, the sample size calculation yielded 68 cases and 136 controls, resulting in a total of 204 participants.

## Sampling procedure

The calculated sample size was allocated proportionally between the two centers, taking into account the total number of ARM patients at TASH and SPHMMC from January 2019 to January 2022 [S1 File]. A total of 287 new patients were treated for ARM at the two centers during the specified period. Since ARM is a rare condition, all the identified cases were taken until the required sample size reached. Two controls were chosen on the same day for every ARM case using a systematic random sampling method.

## Data collection

The data collection tools were adapted from different previous peer-reviewed studies. The first version of the questionnaire was prepared in English (S2 File) and then was translated into Amharic. It was then re-translated to the English language by another person to check its semantic equivalence. Data was collected from mothers of cases and controls using pretested and structured face-to-face, interviewer-administered questionnaires. This was also supplemented with visual observation of the medical records of the child and the mother.

## Study variables

**Dependent variable.**  Anorectal malformation (yes or no).

**Independent variables include.**  The independent variables in our study include: socio demographic characteristics, parity, prior obstetric complication (abortion, still birth or early neonatal death), history of antenatal care (ANC) follow up, pregnancy Intention (planed vs unplanned), family history of birth defect in the first and second degree relatives, history of maternal illnesses during or prior to pregnancy, maternal exposure to medications, history of folic acid intake during the pregnancy, body mass index (BMI) during pregnancy, alcohol ingestion, maternal exposure to pesticides (insecticides, herbicides, or fungicides) during first trimester of pregnancy.

## Data quality assurance

In order to ensure the quality of data, the principal investigator recruited and trained data collectors. Two nurses with a BSc degree with previous experience were assigned to conducted face to face interview with mothers. One BSc degree in public health was assigned to supervise data collectors in the process of data collections. The training included an overview of the research objectives, data collection techniques and the utilization of data collection tools especially on how to interview as well as fill the questionnaires and check lists properly. To ensure the reliability of the questionnaire, a pre-test was conducted on 5% (3 cases and 7 controls) of

the total sample size prior to the actual data collection. Throughout the data collection period, the principal investigator conducted daily reviews and checks of the collected data for any omissions as well as assessing completeness and consistency.

## Statistical analysis

Data were entered using Epi Info version 7 and cleaned and analyzed in SPSS version 26. Following descriptive analysis, tabular presentations including frequency distributions and cross-tabulations were used to present the results. Categorical variables were analyzed using the chi-square ($\chi^2$) test. Fisher's exact test was used when the assumptions for the chi-square test for association were not met. We performed a bivariable logistic regression analysis for each maternal risk factor individually to assess its association with the outcome variable, ARM. The crude odds ratio (COR) and the corresponding 95% confidence interval (CI) for each maternal risk factor were determined and the statistical significance of the association was evaluated using the p-value.

All the maternal risk factors identified as potentially significant in the bivariable analysis (P<0.25) were analyzed using multivariable logistic regression model to estimate the independent effects of each maternal risk factor on odds of ARM, while controlling for potential confounding factors. Adjusted odds ratios (AORs) and the corresponding 95% CIs for each maternal risk factor were determined in the multivariable model. Maternal risk with a p-value less than 0.05 was considered statistically significant factors associated with the occurrence of ARM.

Multiple model fit measures were employed to make a more well-rounded evaluation of the model's goodness of fit. Based on the results of the likelihood ratio test, Hosmer-Lemeshow goodness-of-fit test, and the pseudo R-squared measures, the logistic regression model appears to fit our data reasonably well. Variance inflation factor (VIF) was used to measure multicollinearity. The VIF values in our study were below 10, indicating the absence of concerning multicollinearity.

## Operational definitions

- **Anorectal malformations**:- The absence of an anus or an abnormal anus, with or without a fistula, and they were classified according to the Krickenbeck criteria [7].

- **Alcohol Consumption**: Self-reported intake of alcoholic beverages, including beer, wine, and distilled spirits (e.g., liquor, spirits, hard liquor) with frequency of 1 or more drink per day.

- **Passive smoking**: Regular exposure to tobacco smoke from the smoking of others at home or work place for an extended period (several hours per day or multiple days per week) during pregnancy.

- **Active smoking:** Smoking cigarettes during pregnancy, regardless of the number of cigarettes smoked per day.

- **Exposure to Pesticide**: Any documented or self-reported use, handling, or environmental contact with chemical pesticides including insecticides, herbicides, fungicides, and rodenticides, in a home or occupational setting. Exposure could occur through various routes, including living close to places where pesticides are frequently used and applying, mixing, or spraying pesticides directly.

### Ethical clearance

Ethical approval for the study was obtained from the Department of Research Ethics Review Committee (DRERC) at the Department of Anatomy, School of Medicine, and Addis Ababa University, with reference number DRERC/06/22 dated June 2022. Additionally, approval was granted by the institutional review board (IRB) of St. Paul's Hospital Millennium Medical College, with reference number PM 23/291, dated December 2022. The study procedure and its objectives were disclosed to all participants. Participants were informed that their participation was voluntary and could be withdrawn at any time, and that non-participation would not affect the services they received from the hospital.

Written and signed consent was obtained from all study participants before administering the questionnaires and conducting face-to-face interviews. Participants were not required to write their names on the questionnaires to maintain confidentiality, and no identifying information was recorded. The data were used solely for the purposes of the study to ensure confidentiality.

## Results

### Socio-demographic characteristics of mothers

In this study, a total of 204 (68 cases and 136 controls) mothers were identified and interviewed at the pediatrics surgical units, achieving a response rate of 100%. About 87.5% of mothers for controls and more than half for cases 76.5% were in the age group of 20–34 years. The patients were from different parts of the country, 60.3% for controls and 38.2% for cases came from Addis Ababa, while the remaining from other regions of the country. Regarding educational status of the mothers 35.3% for cases and 39.7% for controls were high school graduates. About 67.6% of mothers for cases and 94.9% of mothers for controls were urban residents (Table 1).

### Reproductive and obstetric history of respondents

In the study, the majority of women were multipara, with 54(79.4%) for cases and 83(61%) for controls. Approximately 44.1% of the pregnancies for the cases were unplanned, compared to 13.2% for the controls. Furthermore, the majority of mothers gave birth at term (37–42 weeks), with 89.7% for cases and 89.7% for controls, while a small percentage gave birth before 37 weeks, with 4.4% for cases and 7.3% for controls.

Around 22.1% of mothers for cases and 16.2% for controls had a history of previous obstetric complications (abortions and still birth). About 13.2% of mothers for cases and 2.9% for controls reported a family history of birth defects. Among all mothers, only a small percentage did not attend antenatal care follow-up, with 3% for controls and 11.8% for cases. About 43% of mothers for the case group and 81.6% for the control group were taking folic acid supplementation during their pregnancy, but only 22% of mothers for cases and 47.1% for controls took folic acid supplementation in the first trimester of pregnancy (Table 2).

### Maternal medical, lifestyle and environmental factors

The majority of mothers had a normal body mass index, with 40(68.4%) for cases and 93(59%) for controls. Approximately 25% of mothers were overweight, with 19(28%) for cases and 33 (24.3%) for controls. In our study maternal medical illness during pregnancy was seen in 44% of mothers for cases and 16.2% of mothers for controls. Out of this the commonest medical illnesses were Urinary tract infection and anemia. Out of the study participants, medication use in first trimester of pregnancy was noticed in 22% of mothers for cases and 5.9% of mothers

**Table 1. Socio-demographic characteristics of mothers included in the study conducted at specific hospitals located in Addis Ababa, Ethiopia, during the year 2022.**

| Characteristics | Case (%) (N = 68),N(%) | Control (%) (N = 136),N(%) |
|---|---|---|
| Mothers age | | |
| 15–20 | 0 | 10 (7.3) |
| 20–34 | 52 (76.5) | 119 (87.5) |
| > 34[a] | 16 (23.5) | 7 (5.2) |
| Mother's Education | | |
| Illiterate | 22 (32.4) | 3 (2.2) |
| Read and write | 3 (4.4) | 14 (10.3) |
| Elementary education | 10 (14.7) | 36 (26.5) |
| High school education | 24 (35.3) | 54 (39.7) |
| Higher education | 9 (13.2) | 29 (21.3) |
| Mother's Occupation | | |
| Housewife | 39 (57.3) | 95 (69.8) |
| Student | 1 (1.5) | 2 (1.6) |
| Government employee | 13 (19.1( | 15 (11) |
| Private employee | 10 (14.7) | 22 (16) |
| Farmer | 5 (7.4) | 2 (1.6) |
| Mother's region | | |
| Addis Ababa | 26 (38.2) | 82 (60.3) |
| Oromia | 30 (44.1) | 33 (24.3) |
| Amhara | 8 (11.8) | 11 (8.1) |
| SNNP | 3 (4.4) | 7 (5.1) |
| Afar | 1 (1.5) | 0 (0) |
| Somalia | 0 (0) | 3 (2.2) |
| Place of Residence | | |
| Urban | 46 (67.6) | 129(94.9) |
| Rural | 22 (32.4) | 7(5.2) |
| Religion | | |
| Orthodox | 37 (54.4) | 78 (57.3) |
| Muslim | 23 (33.8) | 42 (30.9) |
| Protestant | 8 (11.8) | 16 (11.8) |

[a] Advanced maternal age

SNNP, South Nations and Nationalities

for controls. Alcohol consumption during first trimester of pregnancy was reported in about 30.9% of mothers for case and 9.6% for controls. About 5.9% of mothers for cases and 0.7% for controls had history of passive cigarette smoking during their first trimester of pregnancy. In addition, about 17.6% of mothers for cases and 2.9% for controls were exposed to pesticides in the first trimester pregnancy (Table 3).

## Risk factors associated with anorectal malformation

Bivariable Binary logistic regression analyses revealed significant associations between ARM and maternal exposure to various factors during their first trimester of pregnancy. These factors included alcohol consumption, medication use, passive cigarette smoking, maternal medical illness, exposure to pesticides, lack of folic acid supplementation, advanced maternal age

**Table 2. Reproductive and obstetric characteristics of mothers included in the study, conducted at specific hospitals located in Addis Ababa, Ethiopia, during the year 2022.**

| Characteristics | Case (%) | Control(%) |
|---|---|---|
| | (N = 68),N(%) | (N = 136),N(%) |
| Parity | | |
| Multiparous | 54 (79.4) | 83 (61) |
| Primiparous | 14(20.6) | 53 (39) |
| Gestational age at delivery | | |
| Preterm | 3 (4.4) | 10(7.3) |
| Term [a] | 61(89.7) | 122(89.7) |
| Post term | 4(5.9) | 4 (3) |
| ANC follow up | | |
| Yes | 60 (88.2) | 132(97) |
| No | 8 (11.8) | 4 (3) |
| Lack of Folic acid intake during pregnancy | | |
| yes | 53 (77.9) | 72(52.9) |
| No | 15 (22.1) | 64(47.1) |
| Past obstetric complication [b] | | |
| Yes | 15 (22.1) | 8 (16.2) |
| No | 53 (77.9) | 128 (83.8) |
| Unplanned pregnancy | | |
| Yes | 30 (44.1) | 18 (13.2) |
| No | 38 (55.9) | 118 (86.8 |
| Family History of birth defect | | |
| Yes | 9 (13.2) | 4 (2.9) |
| No | 59 (86.8) | 132 (97.1) |

ANC, antenatal care

[a] 37–42 weeks of gestational age

[b] abortion, still birth.

(age >35 years), history of obstetric complication, and family history of birth defects. Factors such as maternal education, occupation, parity and maternal obesity were not found to have a statistically significant association with ARM.

The variables that were found to be statistically significant in the bivariable binary logistic regression were passed into the next level of statistical analysis, multivariable logistic regression. After applying both bivariable and multivariable binary logistic regression only five variables had showed an overall significant association with the development of ARM at 5% level of significance. Accordingly, women who consumed alcohol during the first trimester of pregnancy were 4.7 times more likely to give birth to newborns with ARM compared to those who did not drink alcohol (Adjusted Odds Ratio (AOR) = 4.71; 95% Confidence Interval (CI) = 1.71 to 12.00). Similarly, women who had a history of taking medication for various illnesses during the first trimester of pregnancy had a 5-fold increased likelihood of having newborns with ARM (AOR = 4.86; 95% CI = 1.29 to 18.32).

In addition, advanced maternal age during pregnancy, defined as age over 35 years, was also associated with a 4-fold higher odds of having a newborn with ARM compared to younger mothers (AOR = 4.22; 95% CI = 1.21 to 14.69). Unplanned pregnancy was significantly associated with the occurrence of ARM, with an AOR of 3.70 and a 95% CI of 1.55 to 8.83. Furthermore, the study identified a statistically significant association between family history of birth

**Table 3. Medical, lifestyle, and environmental exposure characteristics of mothers included in the study, conducted at specific hospitals located in Addis Ababa, Ethiopia, during the year 2022.**

| Characteristics | Case (%) | Control (%) |
|---|---|---|
| Maternal medical illness | | |
| Yes | 30 (44.1) | 22 (16.2) |
| No | 38 (55.9) | 114 (83.8) |
| Types of maternal medical illness | | |
| DM | 1(1.5) | 2(1.5) |
| Anemia | 7 (10.3) | 4(2.9) |
| UTI [a] | 12(17.6) | 6(4.4) |
| Epilepsy | 1(1,5) | 0(0) |
| Other [b] | 9(13.2) | 10(7.3) |
| Body mass index (Kg/m$^2$) | | |
| Under weight | 5 (7.3) | 7(5.1) |
| Normal | 40 (59) | 93(68.4) |
| Over weight | 19 (28) | 33(24.3) |
| Obese | 4 (5.7) | 3 (2.2) |
| Medication use in the 1st trimester | | |
| yes | 15(22) | 8(5.9) |
| No | 53(78) | 128(94.1) |
| Alcohol consumption | | |
| Yes | 21(30.9) | 13(9.6) |
| No | 47(69.1) | 123(90.4) |
| Passive smoking in the 1st trimester | | |
| Yes | 4(5.9) | 1(0.7) |
| No | 64(94.1) | 135(99.3) |
| Active smoking | | |
| Yes | 0 | 2(1.5) |
| No | 68 (100) | 134 |
| Exposure to Pesticide in the 1st trimester | | |
| Yes | 12(17.6) | 4(2.9) |
| No | 56(82.4) | 132(97.1) |

[a] Urinary tract infection

[b] Hypertension, HIV, HPV, Asthma, DVT, STI, and Febrile illness

defects and ARM, where mothers with a family history in first and second-degree relatives had 6 times higher odds of having a child with ARM compared to those without such a family history (AOR = 6.15; 95% CI = 1.24 to30.58)(Table 4).

## Discussion

Existing literature suggests that ARM is a multifactorial condition, involving both genetic and environmental factors that play a role in its pathogenesis. This study investigated the association between a number of the maternal factors and the development of ARMs. In the present study, individuals with a family history of birth defects, maternal alcohol consumption, medication use, advanced maternal age and unplanned pregnancy were found to be significantly associated factors with ARM.

**Table 4. Bivariate and multivariate binary logistic regression analysis examining risk factors for ARM among mothers visiting at specific hospitals in Addis Ababa, Ethiopia, during 2022.**

| Characteristics | Case (%) | Control (%) | COD,95% CI | AOD 95% CI | P-value |
|---|---|---|---|---|---|
| ANC follow up | | | | | |
| Yes | 60 (88.2) | 132 (97.1) | 0.227(0.07,0.78) | 0.742(0.12, 4.59) | 0.744 |
| No | 8 (11.8) | 4 (2.9) | 1 | 1 | |
| Advanced maternal age [a] | | | | | |
| Yes | 16 (23.5) | 7 (5.1) | 5.670(2.21,14.58) | 4.22 (1.21, 14.69) | 0.024* |
| No | 52 (76.5) | 129 (94.9) | 1 | 1 | |
| Maternal illness | | | | | |
| Yes | 30 (44.1) | 22 (16.2) | 4.091 (2.11,7.92) | 2.27 (0.89, 5.72) | 0.085 |
| No | 38 (55.9) | 114 (83.8) | 1 | 1 | |
| Medication use | | | | | |
| Yes | 15 (22) | 8 (5.9) | 4.528(1.81, 11.36) | 4.86 (1.29, 18.32) | 0.02* |
| No | 53 (78) | 128 (94.1) | 1 | 1 | |
| Family history of birth defect | | | | | |
| Yes | 9 (13.2) | 4 (2.9) | 5.034 (1.49, 17.00) | 6.15 (1.24, 30.58) | 0.026* |
| No | 59 (86.8) | 132 (97.1) | 1 | 1 | |
| Passive Smoking | | | | | |
| Yes | 4 (5.9) | 1 (0.7) | 8.437 (0.92,77.02) | 6.85 (0.32, 43.39) | 0.292 |
| No | 64 (94.1) | 135 (99.3) | 1 | 1 | |
| Exposure to Pesticide | | | | | |
| Yes | 12 (17.6) | 4 (2.9) | 9.500 (2.58, 34.97) | 2.53 (0.49, 12.89) | 0.265 |
| No | 56 (82.4) | 132 (97.1) | 1 | 1 | |
| Lack Folic acid intake during the 1st trimester | | | | | |
| Yes | 53 (77.9) | 72 (52.9) | 3.141 (1.62, 6.12) | 2.31 (0.98, 5.42) | 0.055 |
| No | 15 (22.1) | 64 (47.1) | 1 | 1 | |
| Unplanned pregnancy | | | | | |
| Yes | 30 (44.1) | 18 (13.2) | 5.175(2.60, 10.31) | 3.70 (1.55, 8.83) | 0.003* |
| No | 38 (55.9) | 118 (86.8 | 1 | 1 | |
| Past obstetric complication | | | | | |
| Yes | 15 (22.1) | 8 (16.2) | 4.528(1.81, 11.32) | 1.44 (.44, 4.72) | 0.545 |
| No | 53 (77.9) | 128 (83.8) | 1 | 1 | |
| Alcohol consumption | | | | | |
| Yes | 21 (30.9) | 13 (9.6) | 4.617 (2.12,10.12) | 4.71 (1.71,12.00) | 0.003* |
| No | 47 (69.1) | 123 (90.4) | 1 | 1 | |

COR, Crude odds ratio; AOR, Adjusted odds ratio; CI, Confidence interval

*indicates statistical significance.

[a] age >34years.

Several environmental factors have been investigated in different studies as risk factors of ARM, but the results have been inconsistent and heterogeneous. In our study, children with a family history of birth defects including ARM in their first and second-degree relatives were more likely to develop ARM than those without a family history. Our results are consistent with research from Germany and the Netherlands that observed an increased risk of ARM in those with a family history of birth abnormalities [11–13].

Our research suggests that maternal alcohol consumption during early pregnancy may increase the risk of ARM, which is consistent with studies conducted in Japan on patients with

ARM [24, 25]. This association may be due to the fact that alcohol exposure can disrupt normal embryonic development processes such as cell proliferation, migration, and differentiation by crossing the placental membrane. However, studies conducted in Germany, the Netherlands, and Sweden did not find a significant association between alcohol consumption and the occurrence of ARM [13, 25, 26]. One possible explanation for these contradictory findings is that the effects of alcohol on embryonic development depend on the dose, timing, duration, and pattern of exposure. Furthermore, differences in cultural alcohol consumption patterns across various communities may have contributed to the observed differences as well.

This study found that medication use during the first trimester of pregnancy is another potential risk factor for ARM. This result aligns with other studies, including a meta-analysis on maternal drug use and ARM occurrence. The meta-analysis discovered links between ARM and various drug categories [26]. A research study conducted in Ethiopia on congenital anomalies also reported this association [24]. However, some reports did not find a significant association between medication use during pregnancy and ARM [11, 13]. These conflicting results in the literature may be due to various factors, such as differences in study design, sample size, and the specific medications investigated.

In our literature search, we were unable to find a study that specifically demonstrated a link between ARM and advanced maternal age. However, many research studies have documented a link between advanced maternal age and the occurrence of other congenital anomalies. We found that both advanced maternal age and unplanned pregnancy were associated with a higher risk of developing ARM. The possible mechanism for the association could be the fact that older mothers have a higher risk of genetic mutations and an increased incidence of aneuploidy, which could potentially contribute to the development of ARM [27–29].

The present study found that women with unplanned pregnancies had a higher likelihood of giving birth to children with ARM compared to those with planned pregnancies. A direct association between ARM and unplanned pregnancy had not been reported previously. However, the existing literature has demonstrated increased risks of other congenital anomalies, such as congenital heart diseases [30], and neural tube defects [31], among pregnancies that were unintended.

The potential reasons for the higher risk of anorectal malformation observed in unplanned pregnancies could be multifaceted. Firstly, unplanned pregnancies may be associated with a lower likelihood of receiving timely and comprehensive prenatal care [32]. Secondly, during an unintended pregnancy, women may be more likely to be exposed to various risk factors that can contribute to the development of ARM, such as certain medications, infections, or environmental exposures. These risk factors may be more prevalent or less controlled in the context of an unplanned pregnancy. Lastly, an unintended pregnancy can be a significant source of stress for the mother. Elevated maternal stress levels have been linked to an increased risk of various congenital abnormalities, including ARM, potentially due to the physiological effects of stress on fetal development [33].

Furthermore, the present study found that several factors, including maternal BMI, maternal medical illness during pregnancy, lack of folic acid intake, and passive smoking, did not have a significant association with the occurrence of ARM. This contrasts with other studies conducted in Germany, the Netherlands, Sweden, China, and Japan, where one or a combination of these factors were found to be associated with ARM, resulting in inconsistent findings [11, 13, 34]. The variation in results across different studies may be due to differences in study design, such as variations in the definition and classification of ARM, sample size, recruitment strategies, and data collection and analysis methods.

Additionally, differences in healthcare practices and access to prenatal care may also contribute to the heterogeneity in the results. For instance, limited access to prenatal screening and diagnostic tests in some regions could affect the detection and reporting of ARM cases.

## Strengths of the study

- Comprehensive investigation of a wide range of potential risk factors for ARMs.

- Employment of trained, experienced health professionals for data collection, and evaluation/ diagnosis of ARMs by pediatric surgeons, ensuring expertise and accuracy.

## Limitations of the study

- As this was a hospital-based study, there is a possibility that the researchers may have missed patients who did not seek medical treatment, potentially introducing selection bias.

- Since the study was not community-based, it may be difficult to extrapolate the conclusions to the target population at large.

- Recall bias may have been introduced, as information about maternal exposure was obtained retrospectively.

- ARMs are relatively rare congenital anomalies, and the sample size in an unmatched case-control study may have been limited, reducing the statistical power to detect significant associations.

## Conclusion

The study findings indicate that anorectal malformation (ARM) risk is substantially elevated for infants with a family history of birth defects, as well as for those whose mothers consumed alcohol or took medications during the first trimester of pregnancy. Additionally, advanced maternal age and unplanned pregnancy emerged as other significant contributing factors. These results underscore the importance of pre-pregnancy and prenatal care to identify and mitigate key risk factors, in order to help prevent the occurrence of this congenital abnormality. The identified risk factors suggest a need for a multifaceted approach addressing genetic, maternal lifestyle, behavioral, and health-related influences to effectively reduce the burden of ARM. Further population-based research is needed to more comprehensively examine the relationship between maternal risk factors and anorectal malformations.

## Supporting information

**S1 File. Sampling procedure.**
(PDF)

**S2 File. Consent form (English and Amharic version).**
(DOCX)

**S3 File. English version questionnaire.**
(PDF)

## Acknowledgments

The researchers would like to express their deepest gratitude to all the staff members of the Department of Pediatric Surgery at Addis Ababa University and St. Paul's Hospital Millennium Medical College for their invaluable support throughout the study. The researchers' gratitude also extends to the trained data collectors at each of the participating hospitals in the study area. Their dedicated efforts in data collection were instrumental in the successful completion of this research project.

## Author Contributions

**Conceptualization:** Samrawit Solomon.

**Data curation:** Samrawit Solomon.

**Formal analysis:** Samrawit Solomon, Fisseha Temesgen, Solomon Tibebu, Hana Abebe, Girma Seyoum.

**Funding acquisition:** Samrawit Solomon.

**Investigation:** Samrawit Solomon, Fisseha Temesgen, Solomon Tibebu, Hana Abebe, Girma Seyoum.

**Methodology:** Samrawit Solomon, Fisseha Temesgen, Solomon Tibebu, Hana Abebe, Girma Seyoum.

**Project administration:** Samrawit Solomon.

**Resources:** Samrawit Solomon, Fisseha Temesgen, Solomon Tibebu, Hana Abebe.

**Software:** Samrawit Solomon, Solomon Tibebu.

**Supervision:** Samrawit Solomon, Fisseha Temesgen, Solomon Tibebu, Hana Abebe, Girma Seyoum.

**Validation:** Samrawit Solomon, Solomon Tibebu, Hana Abebe, Girma Seyoum.

**Visualization:** Samrawit Solomon, Fisseha Temesgen, Solomon Tibebu, Girma Seyoum.

**Writing – original draft:** Samrawit Solomon, Solomon Tibebu, Girma Seyoum.

**Writing – review & editing:** Samrawit Solomon, Solomon Tibebu, Girma Seyoum.

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
