## [Decision Letter · Decision Letter 0]

23 May 2024

PONE-D-24-13124Maternal risk factors associated with Anorectal malformation at selected governmental hospitals in Addis Ababa, Ethiopia, 2023: A case–control studyPLOS ONE

Dear Dr. Tibebu,

Thank you for submitting your manuscript to PLOS ONE. After careful consideration, we feel that it has merit but does not fully meet PLOS ONE’s publication criteria as it currently stands. Therefore, we invite you to submit a revised version of the manuscript that addresses the points raised during the review process.

Thank you for providing the manuscript titled "Maternal risk factors associated with Anorectal malformation at selected governmental hospitals in Addis Ababa, Ethiopia, 2023: A case–control study". As an academic editor, I have the following comments and questions:

**Introduction:**

The introduction provides a good overview of anorectal malformations and the need to assess maternal risk factors. However, it could be strengthened by including more details on the specific knowledge gaps or limitations of previous studies that this study aims to address.It would be helpful to include the research objectives more clearly, perhaps as a separate paragraph at the end of the introduction.

**Methods:**

The study design, setting, and sampling approach are clearly described. It would be good to provide more details on how the sample size was calculated, such as the expected effect size, power, and significance level used.The data collection and analysis methods are well-explained. It would be useful to include information on how anorectal malformations were diagnosed and classified in this study.The statistical analysis plan is comprehensive, but it would be helpful to explain the rationale for using both bivariate and multivariate analyses.

**Results:**

The results are presented clearly and the key findings are highlighted. It would be good to include a table summarizing the demographic and clinical characteristics of the study participants.The interpretation of the results, particularly the adjusted odds ratios, could be strengthened by discussing the plausibility and potential mechanisms underlying the observed associations.

**Discussion:**

The discussion section provides a good synthesis of the key findings and compares them to existing literature. It would be valuable to discuss the strengths and limitations of the study design and how they may have influenced the results.The concluding remarks could be expanded to highlight the public health implications of the findings and propose future research directions.

**Overall:**

The manuscript is generally well-written and organized. Consider improving the flow between sections and ensuring consistent terminology (e.g., "ARM" vs. "anorectal malformation"). Check the formatting and ensure all references are complete and correctly cited. In summary, this is a well-designed and conducted study that contributes to the understanding of maternal risk factors for anorectal malformations in the Ethiopian context. With some revisions to strengthen the introduction, methods, results interpretation, and discussion, this manuscript would be a valuable addition to the literature.

**.** Be sure to:Indicate which changes you require for acceptance versus which changes you recommendAddress any conflicts between the reviews so that it's clear which advice the authors should followProvide specific feedback from your evaluation of the manuscript==============================

We look forward to receiving your revised manuscript.

Kind regards,

Fentahun Adane Nigat, PhD

Academic Editor

PLOS ONE

4. Please amend your list of authors on the manuscript to ensure that each author is linked to an affiliation. Authors’ affiliations should reflect the institution where the work was done (if authors moved subsequently, you can also list the new affiliation stating “current affiliation:….” as necessary)

Additional Editor Comments:

Thank you for providing the manuscript titled "Maternal risk factors associated with Anorectal malformation at selected governmental hospitals in Addis Ababa, Ethiopia, 2023: A case–control study". As an academic editor, I have the following comments and questions:

Introduction:

1. The introduction provides a good overview of anorectal malformations and the need to assess maternal risk factors. However, it could be strengthened by including more details on the specific knowledge gaps or limitations of previous studies that this study aims to address.

2. It would be helpful to include the research objectives more clearly, perhaps as a separate paragraph at the end of the introduction.

Methods:

3. The study design, setting, and sampling approach are clearly described. It would be good to provide more details on how the sample size was calculated, such as the expected effect size, power, and significance level used.

4. The data collection and analysis methods are well-explained. It would be useful to include information on how anorectal malformations were diagnosed and classified in this study.

5. The statistical analysis plan is comprehensive, but it would be helpful to explain the rationale for using both bivariate and multivariate analyses.

Results:

6. The results are presented clearly and the key findings are highlighted. It would be good to include a table summarizing the demographic and clinical characteristics of the study participants.

7. The interpretation of the results, particularly the adjusted odds ratios, could be strengthened by discussing the plausibility and potential mechanisms underlying the observed associations.

Discussion:

8. The discussion section provides a good synthesis of the key findings and compares them to existing literature. It would be valuable to discuss the strengths and limitations of the study design and how they may have influenced the results.

9. The concluding remarks could be expanded to highlight the public health implications of the findings and propose future research directions.

Overall:

The manuscript is generally well-written and organized. Consider improving the flow between sections and ensuring consistent terminology (e.g., "ARM" vs. "anorectal malformation"). Check the formatting and ensure all references are complete and correctly cited. In summary, this is a well-designed and conducted study that contributes to the understanding of maternal risk factors for anorectal malformations in the Ethiopian context. With some revisions to strengthen the introduction, methods, results interpretation, and discussion, this manuscript would be a valuable addition to the literature.

Reviewers' comments:

Reviewer's Responses to Questions

**Comments to the Author**

1. Is the manuscript technically sound, and do the data support the conclusions?

Reviewer #1: Yes

Reviewer #2: Yes

Reviewer #3: Partly

2. Has the statistical analysis been performed appropriately and rigorously? 

Reviewer #1: Yes

Reviewer #2: Yes

Reviewer #3: Yes

3. Have the authors made all data underlying the findings in their manuscript fully available?

Reviewer #1: Yes

Reviewer #2: Yes

Reviewer #3: Yes

4. Is the manuscript presented in an intelligible fashion and written in standard English?

Reviewer #1: Yes

Reviewer #2: No

Reviewer #3: No

5. Review Comments to the Author

Reviewer #1: General: It is a relevant topic. It needs proper editing of language usage and for consistent use of terms, figures and flow of idea and avoid repetitive notes

1. Abstract

Settings: Not proper description of study area in methods section of the manuscript

ODK: It is not mentioned in the body (methods) of the manuscript

2. Methods

a. The methods section needs further reorganization

b. The methods section should clearly show the study area, period, settings, design, sampling technique (controls), variables of interest, data collection procedure, data quality assurance methods and analysis methods and only information included in this section can be reported in the abstract section.

c. Why were other congenital anomalies other than ARM were also excluded? (exclusion criteria should be described well)

d. What does non-ARM related disease mean?

e. Hospital based or population based control selected?

f. Inclusion of children: Is this reflected in the analysis?

g. What is the target population of study? Children with ARM or Mothers?

h. Mention as oral consent in other part of the study?

i. Why was 5% used as non-response mark than the usual 10%?

j. How was data quality control assured?

k. How was variable selection made for inclusion (p-values) in the multivariable analysis

l. How was multicollinearity checked?

m. How was the final model of multivariable regression checked?

3. Results

a. How was the overall response rate?

b. Illness: Did you check separately or in groups for each disease entity for significance? (statistical significance might have been different)

c. What does unspecified medications mean?

d. How was it different with different groups of illness analyzed above? (statistical significance might have been different)

e. Which trimester of pregnancy was of interest in data collection? Was the pre-pregnancy period considered? (mainly for medication, alcohol and smoking)

f. Was maternal smoking status assessed other than passive smoking?

g. Redundant paragraph information

h. Check again, some of the %s do not add up to 100% on tables

i. Check again for chi-square assumption in each cell; some of the cells need regrouping for analysis

4. Discussion

a. Only results addressed in the results section and pertinent factors should be included in the discussion

b. General terms should be avoided like, sampling technique, design etc in the discussion section on result alignment or disparity

c. Conclusion

Re-write focused to the study findings; pertinent to the study results, future suggestions, limitations of the study should be raised here

Reviewer #2: The work you have done is very substantial for evidence based birth defects, particularly in the context of anorectal malformations. However, I believe that certain adjustments are necessary to correct editorial and grammatical issues, as well as specific suggestions in each part.

Reviewer #3: Overall

This manuscript addressed an important topic. Appropriate design was used. The discussions and conclusions provided were based on the findings of the study. However, the manuscript shall address the following concerns. The manuscript lacks page and line numbers. This made it providing comments very difficult. The writing quality of the manuscript should be improved. I recommend language editing. There are several typos and grammar problems. E.g. Typos: “Alcohol conception” and “written concent”. A single sentence was placed as a paragraph in several parts of the manuscript. E.g. “By considering 5% non-responding rate, the calculated sample was 68 cases and 136 controls and a total of 204 sample size (9).” and “The data was entered and analyzed using Statistical Package Social Sciences (SPSS) version 26 software.”

Introduction

The introduction is very brief. Authors would have provided evidence about the magnitude of the problem in the international and local settings. The rational of conducting the study was not well stated.

Methods and materials

Authors shall rewrite the methods section. Authors need to go through all the common subsections of the methods section. The manuscript lacks detailed information on data collection tool and the data collectors. Similarly, it lacks on how the data quality was assured.

The authors shall clearly state at which specific unit/department of the hospitals the cases and controls were recruited.

Variables such as Alcohol consumption, Exposure to Pesticide and Passive smoking need to be operationally defined.

The statistical analysis should have been consistently stated. Authors’ mention of “univariate”, “multivariate”, and “multinomial” are not appropriate for a case control study that used binary logistic regression analysis.

Result

Authors shall rewrite this section. The result was not written in a standard way. Authors report the result in an inconsistent manner. The descriptive and analytic results were reported in a mixed manner. It is better to rewrite the result as the descriptive first, followed by association of ARM with predictor variables. Start the associations first by listing the candidate variables for the multivariable analysis and then interpreting the variables that showed significant associations in the multivariable associations.

While the authors employed odds ratio (OR) as a measure of association, the association result has been interpreted as if it is a relative risk (RR). This mistake is serious and has to be corrected in all of the association interpretations.

While Table 1 includes reproductive and obstetric factors, the table tittle mentions only socio-demographic variables.

Discussion

The strength of this manuscript is its discussion. However, it can still be improved by providing scientific justifications for the associated variables. The last paragraph of the discussion shall contain the strength and limitation of the study.

Conclusion

The first two statements “Several studies were conducted on the risk factors of ARM. However, these studies are inconsistent and heterogeneous.” shall better be removed from the conclusion. Authors shall put recommendations based on their conclusion.

6. PLOS authors have the option to publish the peer review history of their article (what does this mean?). If published, this will include your full peer review and any attached files.

Reviewer #1: No

Reviewer #2: **Yes: **Bickes Wube Sume

Reviewer #3: **Yes: **Mihretu Jegnie

---

## [Author Response · Author response to Decision Letter 0]

16 Jul 2024

AUTHOR’S RESPONSE TO ACADEMIC EDITOR COMMENT

Dear editor

I greatly appreciate your wonderful, inspiring, and motivating comments on our manuscript. We value your insightful and helpful feedback, which we believe will enhance the quality of our work. The following changes have been made in response to your suggestions.

INTRODUCTION

Include more details on the specific knowledge gaps or limitations of previous studies that this study aims to address.

Answer - Included in more detail as commented. The previous research studies performed at Tikur Anbessa Specialized Hospital (TASH) and Jimma University Hospital had solely focused on investigating and quantifying the overall burden the medical condition in question. However, up until this point, no studies have been conducted to explore or identify the potential underlying determinants or associated factors that may contribute to the development and occurrence of this particular condition.

Include the research objectives more clearly, perhaps as a separate paragraph at the end of the introduction.

Answer- Comment accepted. Included in more detail as a separate paragraph at the end of the introduction section

METHODS: 

o The study design, setting, and sampling approach are clearly described. It would be good to provide more details on how the sample size was calculated, such as the expected effect size, power, and significance level used.

 Answer – comment accepted and corrected accordingly 

o The data collection and analysis methods are well-explained. It would be useful to include information on how anorectal malformations were diagnosed and classified in this study.

Answer - The materials and methods section of the paper discusses the process used to diagnose anorectal malformations (ARM) in the study participants. However, as the focus of this research project was not on examining the specific patterns or classifications of the different types of ARM, it may not be necessary to provide a detailed discussion of ARM classification schemes within the methods section, as that information may not be directly relevant to the primary goals of the study. Instead, the introduction section of the paper already included a thorough overview of the most up-to-date classification system for anorectal malformations. 

o The statistical analysis plan is comprehensive, but it would be helpful to explain the rationale for using both bivariate and multivariate analyses.

Answer – comment accepted and corrected accordingly. The rationale for using bivariable binary logistic regression and multivariable logistic regression was discussed in the revised manuscript in detail.

RESULTS

o The results are presented clearly and the key findings are highlighted. It would be good to include a table summarizing the demographic and clinical characteristics of the study participants.

 Answer - Comment accepted. I included one additional table showing the

 Pregnancy and obstetric characteristics of the study participants.

o The interpretation of the results, particularly the adjusted odds ratios, could be strengthened by discussing the plausibility and potential mechanisms underlying the observed associations.

Answer - The strength of association between the dependent and independent variables was discussed with reference to the AOR. The discussion section provides a good synthesis of the key findings and compares them to existing literature.

o It would be valuable to discuss the strengths and limitations of the study design and how they may have influenced the results.

 Answer - Comment accepted. Limitations related to the study design like recall

 Bias and lack of power were included in the revised manuscript. 

o The concluding remarks could be expanded to highlight the public health implications of the findings and propose future research directions.

 Answer - Included as commented 

Overall:

The manuscript is generally well-written and organized. Consider improving the flow between sections and ensuring consistent terminology (e.g., "ARM" vs. "anorectal malformation"). 

Check the formatting and ensure all references are complete and correctly cited. In summary, this is a well-designed and conducted study that contributes to the understanding of maternal risk factors for anorectal malformations in the Ethiopian context. With some revisions to strengthen the introduction, methods, results interpretation, and discussion, this manuscript would be a valuable addition to the literature.

. Be sure to:

• Indicate which changes you require for acceptance versus which changes you recommend

• Address any conflicts between the reviews so that it's clear which advice the authors should follow

• Provide specific feedback from your evaluation of the manuscript.

Feedback 

We thoroughly reviewed and updated the entire manuscript. We carefully considered and incorporated each specific suggestion and comment provided by the manuscript reviewers. We found the feedback from the reviewers to be extremely valuable and beneficial. The reviewer comments significantly improved the overall quality and structure of the final manuscript.

---

## [Decision Letter · Decision Letter 1]

9 Aug 2024

Anorectal malformation and associated maternal factors among Children at Tikur Anbessa Specialized Hospital and St Paul's Hospital Millennium Medical College, Addis Ababa, Ethiopia: Unmatched case–control study

PONE-D-24-13124R1

Dear Dr. Solomon,

We’re pleased to inform you that your manuscript has been judged scientifically suitable for publication and will be formally accepted for publication once it meets all outstanding technical requirements.

Kind regards,

Fentahun Adane Nigat, MSc., PhD

Academic Editor

PLOS ONE

Additional Editor Comments (optional):

Reviewers' comments:

Reviewer's Responses to Questions

**Comments to the Author**

1. If the authors have adequately addressed your comments raised in a previous round of review and you feel that this manuscript is now acceptable for publication, you may indicate that here to bypass the “Comments to the Author” section, enter your conflict of interest statement in the “Confidential to Editor” section, and submit your "Accept" recommendation.

Reviewer #1: All comments have been addressed

Reviewer #2: All comments have been addressed

Reviewer #3: All comments have been addressed

2. Is the manuscript technically sound, and do the data support the conclusions?

Reviewer #1: Yes

Reviewer #2: Yes

Reviewer #3: Yes

3. Has the statistical analysis been performed appropriately and rigorously? 

Reviewer #1: Yes

Reviewer #2: Yes

Reviewer #3: Yes

4. Have the authors made all data underlying the findings in their manuscript fully available?

Reviewer #1: Yes

Reviewer #2: Yes

Reviewer #3: Yes

5. Is the manuscript presented in an intelligible fashion and written in standard English?

Reviewer #1: Yes

Reviewer #2: Yes

Reviewer #3: Yes

6. Review Comments to the Author

Reviewer #1: I appreciate the authors for positively entertaining all the comments suggested in the review process. All comments are included or justified reasonably. However, the article may still benefit from thorough editing in language /grammar.

Thank you!

Reviewer #2: I would like to acknowledge the authors for their rigorous work. My comments and suggestions were addressed fully.

Reviewer #3: (No Response)

7. PLOS authors have the option to publish the peer review history of their article (what does this mean?). If published, this will include your full peer review and any attached files.

Reviewer #1: **Yes: **Megbar Dessalegn

Reviewer #2: **Yes: **Bickes Wube Sume

Reviewer #3: **Yes: **Mihretu Jegnie

---

## [Editor Report · Acceptance letter]

11 Sep 2024

PONE-D-24-13124R1 

PLOS ONE

Dear Dr. Tibebu, 

I'm pleased to inform you that your manuscript has been deemed suitable for publication in PLOS ONE. Congratulations! Your manuscript is now being handed over to our production team.

Kind regards, 

on behalf of

Dr. Fentahun Adane Nigat 

Academic Editor

PLOS ONE